# Effects of a New Soft Skills Metacognition Training Program on Self-Efficacy and Adaptive Performance

**DOI:** 10.3390/bs13030202

**Published:** 2023-02-24

**Authors:** Chantal Joie-La Marle, François Parmentier, Pierre-Louis Weiss, Martin Storme, Todd Lubart, Xavier Borteyrou

**Affiliations:** 1LaPEA, Université Paris Cité and Univ Gustave Eiffel, 92100 Boulogne-Billancourt, France; 2Laboratoire Mémoire Cerveau et Cognition, Université Paris Cité, 71 Avenue Edouard Vaillant, 92100 Boulogne-Billancourt, France; 3CNRS, UMR 9221—LEM—Lille Economie Management, IESEG School of Management, Université Lille, 59000 Lille, France

**Keywords:** soft skills, professional training, metacognition, self-efficacy, adaptive performance

## Abstract

Although soft skills training is called for by many scholars and managers, empirical studies on concrete training programs are scarce and do not always have the methodological rigor that is necessary to draw meaningful conclusions about their impact. In the present research, we investigate the effects of a new soft skills metacognition training program on self-efficacy and adaptive performance. To test these effects, we conducted an experiment with a sample of employees of a large firm (*n* = 180). The experiment included pre- and post-measurements and a control condition. The results suggested that participating in the training led to an increase in soft skills metacognition, self-efficacy, and four dimensions of adaptive performance, compared to a control condition. Mediation analyses suggested that an increase in soft skills metacognition led to an increase in self-efficacy, which led, in turn, to an increase in adaptive performance. Theoretical and practical implications are discussed, as well as limitations.

## 1. Introduction

Soft skills are commonly defined as transversal and non-technical skills [1,2,3]. The non-technical nature of soft skills was highlighted since the emergence of this term [4] and this characteristic has been remarkably consensual ever since) [5,6]. Since the introduction of the concept in the literature, scholars and practitioners have considered soft skills to be an important asset at work and beyond for several reasons [4]. First, soft skills have been shown to contribute to individual health, success, and well-being, as well as collective progress [2,7,8]. In a work context, soft skills are considered as crucial in the context of globalization and change in the work environment. In an ever-changing world, that requires an important plasticity in terms of skills [9,10], the transversality of soft skills is an outstanding asset [3,11,12,13,14]. Nevertheless, soft skills cannot be considered as synonymous with transversal skills that encompass technical skills used in a large range of contexts, jobs and fields, such as IT skills and numeracy [15,16].

Soft skills contribute to performance in the workplace, which is why employers consider them as highly desirable [1,2,3,13,14,17,18,19]. For example, soft skills play a decisive role in managerial outcomes and success attainment, especially when risk management [20] and project management implementations [21,22,23] are involved in the tasks. The major relevance of soft skills as management success factors has been recognized in many work contexts, even in so called “hard skills” domains such as IT [24,25]. They are viewed as particularly essential in human centered management and leadership [26,27]. 

In spite of their importance, soft skills are also considered to be underdeveloped in higher education, creating a gap between employers’ expectations and employees’ acknowledged soft skills [2,3,14,15,17,18,19,28,29,30,31]. The development of skills is thus considered a priority to foster employability [13,14,32,33,34,35]. As soft skills may contribute to many positive outcomes in today’s world, their development and effective implementation appears to be a crucial matter, related to achievement and psychological capital at work. In the present paper, we investigate the effectiveness of a new soft skills training program that targets soft skills metacognition. We investigate the extent to which the program has effects that extend to self-efficacy and adaptive performance. 

### 1.1. The Challenges of Training Soft Skills

Research on the development of training modules to foster soft skills faces important challenges. A first challenge relates to the way soft skills should be trained, notably because of what could be called the “transferability challenge”. Many scholars have expressed skepticism concerning the very possibility of training soft skills. First, the transversal property of soft skills raises questions about the possibility to train these skills as such [36]. For example, some scholars have suggested training soft skills through a specific task or context, to foster transferability to real-life settings [12,37,38]. However, even with contextualized training programs, there remains a doubt about the transferability of soft skills training, because implementing soft skills in real-life conditions can be complex [38,39]. Consequently, soft skills training might not have any observable effect on important work outcomes such as performance.

Another concern is related to the cost of the time of training soft skills. Soft skills training usually takes the form of long-term programs [39,40], and uses experiential modes of training, such as mentoring [37] or virtual reality [41]. Therefore, some scholars raised the issue of balancing costs and benefits of soft skills training [38]. Especially when considering that the effectiveness of such training programs depends on external factors. For example, Cornford [42] and Hurrell et al. [43] found that soft skills performance and development also depends on contextual and organizational determinants, such as autonomy or support; consequently, individual soft skills training may not be a sufficient means to develop soft skills. Short and effective training programs might not be easy to create. 

Other challenges faced by scholars interested in developing soft skills are more methodological. For example, assessing soft skills can be challenging [13,28,38,44,45,46]. The difficulty to assess soft skills has been acknowledged since the origin of the concept [4], and is sometimes considered as what defines soft skills [47,48]. To remedy this difficulty, some scholars have proposed to use behavioral scales and assessment centers [4,49,50]. However, this type of assessment is typically expensive and time-consuming [48]. Instead, soft skills training studies generally rely on ad hoc measures, which do not always have satisfactory psychometric properties [40,51,52,53].

In sum, soft skills training is a subject that deserves to be investigated, due to the growing importance of soft skills in the world of work, their transversal value and the gap between employers’ expectations and employees’ estimated level [2,19,32]. A major goal for enterprises, employees, and universities is to find a relatively easy way to train soft skills that would allow transferability from the training program to specific work contexts in such a way that it has measurable effects on work outcomes [5]. Nevertheless, research on ways to train soft skills remains scarce because of the methodological challenges described above, and when studies exist, they often have methodological limitations such as the absence of measurement before the training [40,52,53] or the absence of a control group to provide a baseline allowing a true measure of the effect of the training program [40,41,51,53,54].

### 1.2. A New Approach to Training Soft Skills

Our aim with this paper is to contribute to bridging the gap in the literature on soft skills training by developing and validating a relatively frugal training program that would have observable effects on important work outcomes such as self-efficacy and adaptive performance. To circumvent the core problem of transferability, we propose to target the development of metacognition concerning soft skills’ existence, definition, implementation, and value [52,55]. In other words, we propose an intervention aimed at developing metacognitive knowledge on soft skills, moving from implicit to explicit knowledge.

Metacognition is a concept introduced by [56] which has since been widely studied in different areas of psychology including experimental and cognitive psychology [57]. Metacognition, defined as “cognition about cognition” [58] is commonly considered as a multidimensional construct divided into knowledge of cognition and regulation of cognition [59]. In the present research, we focus on metacognitive knowledge, which includes declarative knowledge—that is, the ability to formally define a given skill—procedural knowledge—that is, knowing how to implement a given skill—and conditional knowledge—that is, knowing the circumstances under which a skill is useful [60]. Metacognitive knowledge is indeed an essential dimension of metacognition that is closely linked to the activation of skills [61] and feeds the processes of metacognitive regulation [62]. Metacognition malleability has been widely established as well as the possibility to enhance metacognition using training procedures [59,63,64]. Given the impact of metacognition on performance in different professional and educational fields [65,66,67,68], training metacognition appears as a major means to enhance soft skills, particularly when complexity, novelty and adaptability are at stake [69]. 

In the present research, we define soft skills metacognition as metacognitive knowledge about soft skills—that is, one’s knowledge about soft skills definition, the way a person practically implements soft skills, and the relevant circumstances in which the individual decides to implement them. The importance of training soft skills metacognition derives from the implicit dimension of soft skills, which has been highlighted in the prior literature [12,17,30,38]. More precisely, soft skills can be acquired and implemented implicitly, that is, through experience and without requiring any formal teaching [70,71]. This implicit property of soft skills is well documented for different soft skills, such as emotional regulation [72], social cognition [73,74,75,76] decision making, notably in complex and urgent situations [73,77] and emotional intelligence and creativity [78]. As it is the case for other implicit skills [66,78,79,80,81,82], soft skills’ explicitation is proposed to improve performance [14,19,30,31,55]. Indeed, the conscious identification of one’s own soft skills and their understanding appear to be a prerequisite for the development of soft skills [30,52]; enhancing metacognition of soft skills appears to be essential for self-assessment and the implementation of effective self-regulation mechanisms in soft skills development.

A training approach aimed at developing soft skills metacognition would overcome the difficulties currently faced in the soft skills training literature that we listed above. First, it would represent a frugal way to train soft skills, as it could rely on existing experience to support participants’ metacognition. Second, the metacognition of soft skills and their transversality could promote transferability, by convincing participants of their relevance in job contexts as well as the capacity to improve them, as highlighted by a literature review [83]. This could, in turn, constitute a contribution to reduce the soft skills gap: (1) by optimizing the level of implementation of soft skills [19,66,78,81], (2) by valuing them, which could support the motivation to develop them, and by enabling managers and employees to acknowledge more precisely the actual soft skills level at stake in professional missions. The metacognition of soft skills among managers and employees could therefore appear to be a prerequisite for their optimal evaluation and development [43].

In the present research, we aim to conduct an experiment to test the effectiveness of a soft skills training program targeting metacognition. In the experiment, we aim to compare a group of participants who followed a soft skills metacognition training program (experimental condition) with a group of participants who did not (baseline condition). In addition, our design includes a pre- and a post-treatment measurement to control for potential pre-existing differences across conditions. Our main hypothesis is that the soft skills metacognition training increases soft skills metacognition compared to a control condition (H1). To show that the effect of the training module does not only concern soft skills metacognition but also other important work outcomes, we study the effects of our training module on self-efficacy and adaptive performance. 

### 1.3. Effects on Self-Efficacy

Self-efficacy, central to Bandura’s social cognitive theory, refers to a self-appraisal of efficacy, defined as “how well one can execute courses of action required to deal with prospective situations” [84]. Self-efficacy has been primarily conceptualized as specific to a given task; nevertheless, Bandura noted that “once established, enhanced self-efficacy tends to generalize to other situations” [85,86,87,88]. General self-efficacy is now used in numerous research studies in various fields [89,90,91,92,93], and its relevance has been particularly observed at work [94], as well as activities requiring multiple behaviors simultaneously, which is the case of complex tasks involving adaptation [89]. For both these reasons, training targeting self-efficacy has been developed in the literature and appears as a key variable in employment [95].

We propose that the development of metacognition and particularly metacognitive knowledge of soft skills may enhance self-efficacy for different reasons. First, because soft skills are often implicit, the way they contribute to performance is poorly known by lay people [73,86,96,97]; thus, raising metacognition about one’s soft skills could increase the perception of one’s capacities and, consequently, self-efficacy [66,69,78,85,86,94,98,99,100]. Second, because soft skills are transversal, their mastery can be useful in many situations, which should lead to an increase in general self-efficacy [14,85]. Consequently, we hypothesize that training soft skills metacognition will lead to an increase in self-efficacy—mediated by post-training soft skills metacognition—compared to a control condition (H2).

### 1.4. Effects on Adaptive Performance

Adaptation is considered as an essential skill in the 21st century, characterized as Volatile, Uncertain, Complex, and Ambiguous (VUCA), [101,102]. Adaptability and its link with performance at work have given rise to a growing literature in the previous decades [103,104]. In the wake of change accelerations in firms’ environment, employees are required to cope with resistances, difficulties, and stress induced by transformations. Progressively, tools and concepts have flourished to describe, conceptualize, and assess associated skills such as agility [105,106], individual and behavioral adaptability [107], managerial coping [108], and openness to changes [109]. Among concepts derived from research on organizational and individual adaptation, adaptive performance appears to be among the most scrutinized [110,111,112,113,114]. This concept, investigated extensively in the literature [110,111,112,113,114,115,116,117], can be defined as “the proficiency with which an individual alters his or her behavior in response to the demands of a new task, event, situation, or environmental constraints” [118]. 

Adaptive performance has a strong relationship with coping, as it contributes to adaptation and resilience when faced with novelty [107,113,118]. However, it is more related with the world of work, because it considers behaviors as factors of performance [110,113]. As a result, adaptive performance is related to general performance and productivity, both at the individual and the collective levels [103,104,107,113], and to other positive outcomes, such as job satisfaction, lifelong learning, teamwork, customer-oriented practices, etc. [107,115,118,119,120].

Some observations may suggest a direct link between raising metacognition on soft skills and the increase of adaptive performance. Indeed, soft skills are often mentioned in research on adaptive performance, and named “adaptive skills” [2,13,19]. Conversely, adaptive performance includes the use of several skills, which are considered as soft skills, including stressful situation management, creative problem solving, tolerance of uncertainty and unpredictability, and interpersonal skills [8,115,118,121]. Following the impact of metacognition on performance as previously developed, the increase of soft skills metacognition can contribute to the development of adaptive performance [19,122], by enhancing the performance of soft skills implementation.

Additionally, soft skills metacognition may contribute to adaptive performance by enlarging soft skills flexible regulation facing novel and unpredictable contexts. Metacognition about soft skills indeed focuses people’s attention and interest on implicit behaviors and reactions linked with soft skills, enlarging their possible regulation [123,124]. Explicit regulation of implicit mechanisms has been shown to be particularly important for adaptive control of behavior [125] to increase flexible responses to novel and non-routine tasks [126]. Increasing consciousness of implicit processes at stake with soft skills, especially socio-affective soft skills linked with social cognition, optimizes their mode of expression in unexpected situations [73] and decision making [125]. Thus, we hypothesize that training soft skills metacognition will lead to an increase in adaptive performance through induced soft skills metacognition increase, compared to a control condition (H3).

Finally, self-efficacy has a strong impact on adaptive performance, which is potentially causal, as proposed in a recent literature review [117]. Consistent with this prediction, general self-efficacy has been shown to be related to coping at work [108,127,128], which is conceptually close to some dimensions of adaptive performance, such as handling emergencies and crisis situations [107], successful adaptation to uncertain and unpredictable situations [89]. Given this relationship between self-efficacy and adaptive performance and the fact that raising metacognition on soft skills may have a positive impact on self-efficacy, as explained before, we can hypothesize that the effect of raising metacognition about soft skills on adaptive performance will be mediated by self-efficacy. We hypothesize that training soft skills will lead to an increase in adaptive performance compared to a control condition, and that this effect is mediated by self-efficacy (H4).

### 1.5. Model Summary

The relationships between our variables and our hypothesis are summarized in the following model (Figure 1 below).

## 2. Method

### 2.1. Participants

Participants were 180 employees of an international railway group involved in a global organizational transformation during the research period. Among them, 148 attended a training which consisted of a workshop concerning soft skills, and 32 took part in the baseline group described below. All job categories and managerial levels were represented. Participants had diverse types of occupations: train driving and trains driving management, infrastructure building or engineering, passenger flow management, human resources functions and management, central management, etc. Participants were recruited mostly through voluntary participation and due to organizational operational constraints, were not randomly assigned to the various experimental conditions. Our design must thus be considered as a quasi-experiment. Note that the two groups were comparable in terms of demographic characteristics, as well as prior soft skills metacognition, self-efficacy, and adaptive performance, which ensures inter-group comparability (see Table 1).

### 2.2. Experimental Design and Procedure

As this research examines the effect of a soft skills metacognition training module on a series of psychological constructs, we designed two conditions to highlight this effect (see Table 2 for an overview of the two conditions). The first one (referred to as “Baseline condition”) is a set of diverse corporate meetings, which can be considered as usual professional activity. More specifically, the Baseline condition consisted of corporate meetings organized in the railway company involving interpersonal interactions in small groups, and discussions about hard skills of different entities of the firm. They can be considered as usual professional activities within this firm and their themes varied. They always linked the participants’ professional experience with strategic issues for this railway group. As a result, this condition appears to be very similar to the soft skills metacognition training in terms of structure, length, and solicitation of self-efficacy determinants. 

The second condition (referred to as “Soft skills metacognition training”) was a single soft skills workshop designed to optimally develop metacognition on soft skills. A 2-h single workshop provided feasibility of data collection although employees were very busy with their operational functions in a transformational context. It also ensured the attribution of the measured effect to our intervention: the impact of a training consisting of several workshops could also have differed according to participants’ very heterogeneous work contexts.

Soft skills workshop facilitators followed a detailed, standardized set of instructions to lead the different sequences. The soft skills metacognition training was presented as a soft skills workshop aimed at developing knowledge and acknowledgement of these skills, in relation to their essential contribution to adaptation in the workplace. The soft skills training established the explicit value of soft skills execution in adaptive contexts. The beginning of this training focused on a specific range of soft skills chosen to fit with the participants’ transformational context and the culture of the firm. The chosen soft skills were cognitive ones, such as intuitive and divergent thinking, conative ones, such as ambiguity tolerance and perseverance, interpersonal ones, such as communication and collaboration, and emotional ones, such as empathy or stress regulation. The selected set of soft skills and their formal definitions were presented. This sequence therefore started focusing on the descriptive part of metacognitive knowledge concerning soft skills. To develop trainees’ conditional knowledge of soft skills in adaptive circumstances, we asked them to participate in a serious game where they had to perform collectively a contextual recognition of the different soft skills at stake in a story. The third sequence of the training consisted of a quiz dedicated to deepening the conditional knowledge of soft skills and addressing the procedural dimension of soft skills metacognitive knowledge. Participants were then required to link their actual professional experience to the panel of soft skills examined in the workshop in a pitch, to focus on the mastery of the three dimensions of soft skills metacognitive knowledge, specifically on the precise awareness of the circumstances and modalities in which soft skills are used in each person’s professional tasks. The material used included a booklet defining the soft skills discussed, and cards showing their definitions and displayed on a tray to facilitate pedagogical visualization by all participants at any time during the workshop. These soft skills cards were also used to support the participants’ vote at the end of the workshop for the soft skills identified in their colleagues’ pitch; tokens were provided to the players so that they could carry out this vote by placing them on the different cards. The workshop material also includes quiz cards that are distributed in the first phase of soft skills recognition.

Participants who benefited from the soft skills metacognition training were clustered in 12 different soft skills workshop sessions, meaning that 15 participants took part in each session on average. These workshops have been homogenized by implementing a training process for the facilitators, governed by a 4-step process: first the future facilitator takes part in the workshop as a participant, then he/she attends the workshop as an observer of a senior facilitator, participates in the animation of the workshop in pair with a senior facilitator and finally conducts the workshop under the supervision of a senior facilitator.

Regardless of the condition, participants gave their consent and were asked to complete an anonymous self-administered paper-and-pencil questionnaire, just before and just after their participation. The questionnaires included the assessment of variables reported in the current study, as well as other data not reported here but available in an open repository available at https://osf.io/3s4bm/files/ (accessed on 21 January 2023). 

### 2.3. Measures

**Soft skills metacognition scale.** We assessed the development of metacognition on soft skills using a purposely designed scale. The scale comprises 3 × 5 items, measuring the 3 dimensions of metacognitive knowledge (declarative, conditional, procedural) [59,60] for 5 different soft skills, chosen for their functional diversity, their importance in the professional activities and their different degree of prior metacognitive mastery, as measured on a panel of trainees during the first sessions of the soft skills metacognition training: communication (interpersonal skill—high prior metacognitive knowledge), mental flexibility (cognitive skill—low prior metacognitive knowledge), ambiguity tolerance (cognitive skill—low prior metacognitive knowledge), openness (conative skill—high prior metacognitive knowledge), cognitive empathy (emotional skill—average prior metacognitive knowledge) [129]. For declarative metacognition, the item used was “I know when and how to use [the soft skill X]”; for procedural, “I know how to implement [the skill]”; for conditional, “I know when and why use [the skill]”. Participants responded on a four-point rating scale ranging from 1 (strongly disagree) to 4 (strongly agree). The observed scores range between 1 and 4.

The structure of responses showed that: (1) for each soft skill, measures of procedural, conditional, and descriptive metacognition were strongly inter-correlated; (2) the global ratings of metacognition for each soft skill were strongly inter-correlated. These observations thus suggest a hierarchical structure with a unique second-order factor; this model was tested and appeared to fitF data, whereas more complex models produced Heywood cases: χ^2^(170) = 95.54, CFI = 0.95, TLI = 0.94, SRMR = 0.05, RMSEA = 0.05. Internal consistency was also satisfactory with an observed Cronbach’s α of 0.92. We therefore considered that this scale allows us to obtain a unique soft skills metacognition score.

Validity studies were conducted to support the use of this new scale. First, a concurrent and discriminant validity study was run, based on 70 participants. The analysis revealed moderate correlations between our metacognitive subscales (declarative, conditional, procedural) and the Metacognitive Awareness Inventory [60], respectively, Spearman’s ρ = 0.29, *p* = 0.02; ρ = 0.24, *p* = 0.04 and ρ = 0.32, *p* < 0.01. We also measured the link between each soft skill subscale in our soft skills metacognition scale and a psychometric test measuring its auto-evaluated performance, as metacognition in a given domain is correlated to performance in the same domain [100,130]. More specifically, we found that, as expected, the Communication subscale was correlated with the Workplace communication scale [131], ρ = 0.43, *p* < 0.001. The Mental Flexibility subscale did not significantly correlate with the Cognitive Flexibility Inventory [132], ρ = 0.08, *p* = 0.53. This result can be explained by the specificity of mental flexibility inventories, that weakly correlate with neuropsychological approaches to mental flexibility. This could suggest that mental flexibility includes dimensions that are apprehended differently by the scales that measure this variable [133]. Finally, there was a correlation between the Empathy subscale of the metacognition scale and the Empathy subscale of the Interpersonal Reactivity Index [134], ρ = 0.44, *p* < 0.001.

Second, a study of content validity involving six researchers in cognitive and work psychology revealed a good adequation and formulation of items to measure the 3 facets of metacognition. Concerning the selected skills panel, two of them were not deemed essential by experts: openness and tolerance of ambiguity. This fact induces the decision to discard these two skills from the global soft skills metacognition score. The other skills were all judged important or useful to detect metacognition on soft skills, which leads us to keep them as relevant items. 

Measurement invariance checks showed a time and experimental condition configural invariance for the resulting scale after openness and tolerance of ambiguity discard.

Considering all these elements, we can conclude that our soft skills metacognition scale has satisfactory psychometric properties.

**Self-efficacy.** We assessed self-efficacy before and after the condition using a self-reported self-efficacy scale. The General Self-Efficacy scale is a ten-item scale that assesses generalized self-efficacy, especially regarding adaptation and coping abilities [89]. Due to comprehension issues with the existing French version of the General Self-Efficacy Scale, the original English version was retranslated in French using a back translation process. A sample item was “Thanks to my resourcefulness, I know how to handle unforeseen situations”. Participants responded on a four-point rating scale ranging from 1 (totally untrue) to 4 (totally true) to the items of the General Self-Efficacy Scale. The observed scores range between 1 and 4.

We checked the structural validity of the self-efficacy scale using a Confirmatory Factor Analysis (CFA), following the recommendations of Schumacker and Lomax [135]. The CFA revealed a good monofactorial structure: χ^2^(70) = 59.10, CFI = 0.95, TLI = 0.93, SRMR = 0.05, RMSEA = 0.06. The reliability was satisfactory with an observed Cronbach’s α of 0.85. 

We also checked for measurement invariance checks based on the method of Satorra and Bentler [136]. We found a strict time and condition invariance for this scale, when comparing the factor structure between measurements before and after the session, in the soft skills metacognition training and in the baseline condition. This method of measurement invariance compares the fit of diverse models in a confirmatory factor analysis across diverse groups (here, before and after the session, as well as experimental and baseline conditions). The first degree of measurement invariance, called configural, is verified when the four groups have the same factorial structure. The second degree, metric invariance, adds to the latter model the equality of item loadings across groups; this model did not show any significant decrease in fit indices, as compared to the former. The third degree, scalar invariance, is based on a model constraining both items’ loadings and intercepts; this third model did not display a significant change in fit. Finally, strict measurement invariance was verified by showing the good fit of a model constraining items loadings, intercepts, and residuals across groups. Thus, each step of measurement invariance (configural, metric, scalar, strict) was validated through the observation of fit indices on corresponding constrained models, which was not significantly impaired at each step. This means that the self-efficacy scores derived from the scale can be compared across time and conditions.

**Adaptive performance.** We assessed adaptive performance using the French version of the Adaptive Performance Scale [113]. The adaptive performance scale is composed of 19 items measuring five dimensions of behavioral adjustment to work conditions or new situations. A sample item was “I do not hesitate to go against established ideas to propose an innovative solution”. Participants responded on a seven-point rating scale ranging from 1 (strongly disagree) to 7 (strongly agree). The observed scores range between 2 and 7. We used the same criteria of CFA to assess the factor validity of this scale, which appeared to be good: χ^2^(284) = 271.42, CFI = 0.89, TLI = 0.87, SRMR = 0.07, RMSEA = 0.06. Internal consistency was also calculated for each subscale and was found to be satisfactory with observed Cronbach’s α of 0.81, 0.79, 0.79, 0.83, 0.77, for, respectively, creativity, reactivity, training and learning effort, interpersonal adaptability, and managing work stress. 

Measurement Invariance checks showed a strict time and experimental condition invariance for this scale, using the procedure described previously. Additionally, due to the proximity of self-efficacy and adaptive performance items, a confirmatory factor analysis was run to check that the scales load on two separate sets of factors. It resulted in good fit indexes: χ^2^(724) = 585.50, CFI = 0.93, TLI = 0.92, SRMR = 0.06, RMSEA = 0.04.

### 2.4. Data Analytic Strategy

For our main analyses, we relied on the Analysis of Covariance (ANCOVA) approach. In this framework, the post-measurement of the variable of interest is regressed on both the treatment variable (baseline vs. treatment) and the pre-measurement of the variable of interest. As soft skills metacognition workshops, though standardized, were facilitated by different persons, our analyses considered the effect of this clustering by modeling random effects, using linear mixed models. Outliers were detected using Q-Q plots and several tests of the “DHARMa” R package [137]. Using this package, we checked model assumptions such as the normality of residuals, the homoscedasticity through graphical exploration (as no test was available for linear mixed models), the independence of residuals using a Durbin–Watson test (“car” R package) [138], and the absence of multicollinearity, using VIF from the “car” R package. Finally, post-hoc power analyses were conducted, using “simr” R package [139] and showed good power (>80%), except when indicated. 

To increase the power of our analyses, data imputation was performed using R “missForest” package (v. 1.4), which uses random forest algorithms to estimate missing data [140]. The number of missing observations for each variable ranged between 0 and 8 (i.e., 4.44%), globally resulting in less than 1% of missing data. After imputation, the variable-wise out-of-bag (OOB) error was controlled, to ensure estimation reliability. The out-of-bag error corresponds to the prediction error of the algorithm, as evaluated on observations that have not been used to train it: it is thus a measure of the quality of imputation. Self-efficacy imputations showed a low error rate (0.16 < MSE < 0.36) and an acceptable for adaptive performance (0.52 < MSE < 0.88); conversely, age was more uncertain (MSE = 49.66) as well as tenure (MSE = 49.31)—they were consequently not imputed. This manipulation led to the imputation of less than 1% of total data, in line with recommendations of the literature [141].

Data were analyzed using R for all analyses (R v. 4.0.3; RStudio V. 1.4.1103). Additionally, Linear Mixed Effect Models were used, due to the nested design of this experiment—“lmerTest” R package [142]. Data and analysis code are available at https://osf.io/3s4bm/files/ (accessed on 21 January 2023).

## 3. Results

Descriptive statistics are reported in Table 3. 

### Main Analyses

We tested first the effect of the training program on the development of soft skills metacognition (H1). Consistent with the ANCOVA approach, we regressed post-condition soft skills metacognition on the dummy-coded training (vs. baseline) variable and on pre-condition soft skills metacognition. Random intercepts and slopes were modelled for each workshop series number because participants were nested in different workshops facilitated by different people. In line with H1, we found a positive effect of our training program on post-condition soft skills metacognition, t(3.01) = 6.73, b = 0.76, *p* = 0.006, suggesting that the training program increased soft skills metacognition.

Then, we proceeded to test the effect of the training program on the development of self-efficacy (H2). For this analysis, we regressed post-condition self-efficacy on the dummy coded training (vs. baseline) variable and on pre-condition self-efficacy. Random intercepts and slopes were modelled to account for possible variability across workshop groups. In line with H2, we found a positive effect of our training program on post-condition self-efficacy, t(139.23) = 4.50, b = 0.21, *p* < 0.001, suggesting that the training program increased the level of self-efficacy of participants.

We relied on the three-step joint-significance method described by MacKinnon et al. [143] and Yzerbyt et al. [144] to test whether soft skills metacognition mediates the effect of the training on self-efficacy. Consistent with our mediation hypothesis, we found that (1) the training predicted post-condition self-efficacy, (2) that the training predicted post-condition soft skills metacognition, (3) introducing post-condition soft skills metacognition in the model predicting that post-condition self-efficacy reduces the effect of training, although the effect remained significant, t(150.99) = 2.13, b = 0.13, *p* = 0.03. This suggests that the mediation is partial, which partially supports H2.

With our third and fourth hypotheses, we expected an effect of the training program on the development of adaptive performance. For this analysis, we regressed each dimension of post-condition adaptive performance on the dummy coded training (vs. baseline) variable and on each dimension of pre-condition adaptive performance. Like in previous analyses, random intercepts and slopes were modelled to account for possible variability across workshop groups. The results are summarized in Table 4. Consistent with H3 and H4, we found positive effects of soft skills metacognition training on each dimension of adaptive performance, except for the Stress Management dimension where the effect was positive but only marginally significant. 

Note that we did not find support for the mediating role of soft skills metacognition in the effect of the training on adaptive performance (H3). Indeed, introducing post-condition soft skills metacognition in the model predicting post-condition adaptive performance did not reduce the effect of training for any of the dimensions of adaptive performance. Instead, we found that self-efficacy played a mediating role in the effect of the training on adaptive performance (H4). These results are summarized in Table 5. The three-step joint-significance approach suggested that introducing post-condition self-efficacy reduced the effect of the training on each dimension of adaptive performance. This finding supports H4 in which we hypothesized that self-efficacy would mediate the effect of the training on adaptive performance. 

## 4. Discussion

Despite a consensus on the importance of fostering soft skills [3,13], there is still debate on the best way to do it [1,38,45]. We tested with a quasi-experimental design an approach that targets metacognition about soft skills. Most of our hypotheses were supported by the data. First, the soft skills metacognition training led to a significant increase in soft skills metacognition. This finding suggests that it is possible to raise awareness about soft skills in daily professional missions [28,38]. Furthermore, the effect of the training module extended to self-efficacy and adaptive performance. This means that when the knowledge about soft skills moves from tacit to conscious, it can contribute to improve performance and development, especially in complex and volatile environments [78,145]. 

Our findings also suggest that training soft skills metacognition leads to increase in participants’ self-efficacy. The evolution of self-efficacy observed in the study is consistent with the fact that self-efficacy is a dynamic construct with a potential for short-term evolution, which can evolve “as new information and experience are acquired (sometimes during actual task performance)” [146]. This is also consistent with previous research showing that self-efficacy generally rises in training contexts [147,148,149], specifically research reporting single workshop effects [150] and metacognition training [151]. The results of our study support the idea that although self-efficacy is influenced by one’s performance, “people are influenced more by how they read their performance successes than by the successes per se” [84], especially concerning complex task performance [85]. Finally, it contradicts the idea according to which soft skills training has a limited influence on self-efficacy, as some scholars have argued before [38]; instead, it shows that metacognition concerning soft skills has a strong impact on general self-efficacy.

Moreover, we found that our training module impacts four dimensions of adaptive performance as well (Creative Problem Solving, Reactivity, and Interpersonal Adaptability, Training and Learning), as conceptualized by Charbonnier–Voirin and Roussel (113). For each of these dimensions, self-efficacy seems to mediate the effect of the training strongly but partially. This is consistent with the close association between self-efficacy and adaptive performance that has been described in prior studies [108,117,152]. Enhancing self-efficacy may therefore raise at least some dimensions of adaptive performance, that are indeed seen as connected to major soft skills [129,153,154].

Some of our findings were not expected and need to be discussed. First, the fifth dimension of adaptive performance (Working stress management) did not seem to be impacted by the training. This could indicate a specificity of this dimension; for example, it could be more predicted than other dimensions by non-metacognitive determinants, such as trait emotional stability [155]. Second, the model testing the mediation between experimental condition, soft skills metacognition, and adaptive performance failed to reach significance. It suggests that an important part of the training effect on adaptive performance is not mediated by soft skills metacognition, but by other constructs, such as self-efficacy. It can be explained by the effect of the training on other general self-efficacy predictors, such as the ability to deal with corporate transformation, the vicarious experience acquired during the training session, etc. Finally, the relationship between metacognitive knowledge and feeling of competency in a specific domain (here, adaptation) is not necessarily linear, as emphasized by the Dunning–Kruger effect [156,157], which may also explain why more power is required to detect this effect.

### 4.1. Implications

Several practical implications can arise from this study, in the scope of human resource challenges to design effective soft skills training programs. First, this study contributes to the debate about the development of soft skills [38] and the complex issue of soft skills assessment [1]. More specifically, our findings suggest that soft skills metacognition is a promising approach to training soft skills, but also that it can be measured by the assessment that we have developed and validated. 

Regarding the impact of soft skills metacognition on self-efficacy, one should first remember the positive effects of self-efficacy in the professional world as “self-efficacy can enhance or impair performance through their effects on cognitive, affective, or motivational intervening processes” [158]. Earlier studies demonstrated a positive relationship between self-efficacy and other variables contributing to organizational performance [147,159] such as entrepreneurship [160,161,162], adaptability to new technology, innovation (Newman et al., 2018), engagement, leadership [163], productivity and management performance [164,165], complex interpersonal tasks, [146,166] and socialization adjustment [167]. Consequently, training soft skills metacognition could prove to be a frugal and effective way of sustaining performance, engagement, and well-being, by increasing self-efficacy. Our findings about self-efficacy are even more interesting that the literature on the effect of soft skills training on self-efficacy yielded opposite results with less reliable methods [38,168]. 

Furthermore, the observed increase in self-efficacy after the training might even boost the training effects in the long run, as self-efficacy has been shown to be positively related to transfer intentions [169,170,171]. Enhancing the metacognition of soft skills may induce behavioral change of employees and managers. The identification, assessment and sustainability of this change may warrant more research in the future. Finally, in the framework of social-cognitive theory, an increase in self-efficacy is a major determinant of performance, because it impacts the goals that people set for themselves, as well as their motivation to reach them, also when the circumstances are challenging [85,172,173]. More research is needed to investigate how our training module might affect performance in the long run. 

A concrete implementation of this work could be to propose a soft skills awareness workshop to job seekers who are a population whose self-efficacy particularly needs to be supported and who also need to be able to identify and value their valued skills in terms of employability [149]. Soft skills metacognition workshops gathering employers and employees could also be used to reduce the soft skills gap. It would indeed enable employers to identify better, recognize, and describe the soft skills they expect, and it would allow employees to have the skills they already use recognized and to identify those they need to develop if they are expected. Deploying soft skills metacognition workshops among employees could also reduce the soft skills gap by allowing employees to implement them in an optimal way [174]. The positive link between awareness of skills and level of implementation of those skills is indeed developed in numerous works [19,66,78,81]. Organizing soft skills metacognition workshops in the educational and academic field could also help to reduce the soft skills gap by teaching students to identify and value their soft skills. Soft skills metacognition training can help to foster soft skills implementation by identifying those who are more solicited at the individual or team level and help to specify useful additional training for those skills that are highly solicited. Soft skills metacognition training could also be used to understand and reduce ill-being at work by identifying the soft skills that are not sufficiently stimulated and those that are highly stimulated in a given position. Extending the soft skills metacognition workshops by proposing to implement this metacognition in problem solving on critical situations could allow teams to improve their performance by identifying and mobilizing their soft skills. 

### 4.2. Limitations

Our study has many limitations that should be addressed in future research. First, there are limitations related to our sample. Organizational constraints did not allow us to assign randomly each participant to a condition; as a result, our study design must be regarded as a quasi-experiment. Our analyses revealed that the samples in the two conditions were comparable on many demographic variables as well as on our main study variables. However, there might be factors that we have not measured that played a role and could provide alternative explanations to our findings. A replication with random assignment would allow this possibility to be ruled out. In addition, the sample of this study comprised employees from one firm only, and the panel of soft skills chosen was adapted to their cultural framework, which can reduce the generalizability of the results. To be sure that the results of this study can be generalized, further research should collect data from different types of organizations (field of activity, public and private sector, size, etc.). Finally, a free association pilot survey indicated that explicit knowledge about soft skills was very low within the studied population. Further studies should assess the impact of soft skills metacognition training in different populations, controlling the level of metacognitive knowledge before the training [124].

Another important limitation concerns pre- and post-treatment measurements. Specifically, we assessed self-efficacy related to adaptation tasks through a general self-efficacy assessment. Dealing with complex tasks such as adaptation, the relationship between self-efficacy and performance can show reduced precision if self-efficacy assessment does not investigate all the dimensions of the task [92]. A specific and multidimensional adaptation self-efficacy scale could therefore be created, and the relationship between this specific adaptation self-efficacy and general self-efficacy could also be investigated in further studies [150]. In the same way, a specific measure of self-efficacy related to soft skills seems a promising perspective to understand better the effect of this soft skills metacognition training, as shown by some preliminary results. Additionally, the self-efficacy growth induced by a single soft skills metacognition training was assessed right after the session; the continued, long-term impact of the training should be investigated through a cohort/longitudinal study. Finally, despite validity checks of our scale concerning the metacognition of soft skills, further analyses could verify the robustness of the scale we designed for this study before it is used by other researchers when testing the efficiency of soft skills metacognition training programs. 

Finally, the fact that this study was based on a single soft skills metacognition workshop constitutes a methodological limitation. This work should therefore be extended by investigating training including several workshops and controlling the characteristics of the professional context during the training period.

### 4.3. Conclusions

This work contributes to clarify the links between soft skills and performance by showing how soft skills metacognition has an impact on self-efficacy and adaptive performance in a transformational context. It paves the way to numerous practical applications in the workplace to reduce the soft skills gap and foster soft skills acknowledgement as well as implementation level. Further research should investigate the generalization of results in different types of organizations and transformations. This work should also be pursued by identifying which methodological protocol optimizes the impact of soft skills metacognition training according to the number of workshops included in the training and work condition characteristics. The sustainability of soft skills metacognition impact highlighted by this work should also be investigated.

## Figures and Tables

**Figure 1 behavsci-13-00202-f001:**
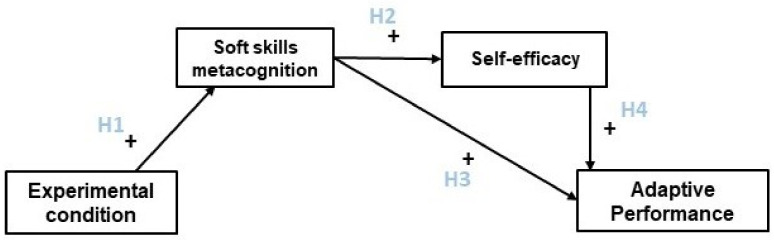
Model for a Soft Skills Awareness Intervention.

**Table 1 behavsci-13-00202-t001:** Between-groups comparison for demographic variables and prior levels of soft skills metacognition, self-efficacy and adaptive performance.

	Baseline Group *n* = 32	Soft Skills Metacognition Training Group *n* = 148	Kruskal-Wallis Test
Age	Average: 42.28 SD: 7.67 NA: 0	Average: 40.68 SD: 8.68 NA: 2	*χ^2^* (1) = 1.19 *p* = 0.28
Tenure	Average: 14.68 SD: 7.21 NA: 1	Average: 15.31 SD: 7.91 NA: 8	*χ^2^* (1) = 0.24 *p* = 0.88
Manager	Manag.: 19 Non-man.: 13 NA: 0	Manag.: 93 Non-man.: 52 NA: 3	*χ^2^* (1) = 0.26 *p* = 0.61
Sex	F: 14 M: 18 NA: 0	F: 84 M: 64 NA: 0	*χ^2^* (1) = 1.75 *p* = 0.18
Prior self-efficacy	Average: 3.31 SD: 0.33 NA: 0	Average: 3.22 SD: 0.37 NA: 0	*χ^2^* (1) = 1.50 *p* = 0.22
Prior soft skills metacognition	Average: 2.49 SD: 0.57 NA: 0	Average: 2.29 SD: 0.65 NA: 0	*χ^2^* (1) = 2.43 *p* = 0.12
Prior adaptive Performance—Creativity	Average: 5.31 SD: 0.81 NA: 0	Average: 5.14 SD: 0.91 NA: 0	*χ^2^* (1) = 0.81 *p* = 0.37
Prior adaptive Performance—Reactivity	Average: 5.31 SD: 0.79 NA: 0	Average: 5.21 SD: 0.80 NA: 0	*χ^2^* (1) = 1.27 *p* = 0.26
Prior adaptive Performance—Training and learning effort	Average: 5.91 SD: 0.79 NA: 0	Average: 5.76 SD: 0.75 NA: 0	*χ^2^* (1) = 1.56 *p* = 0.21
Prior adaptive Performance—Interpersonal Adaptability	Average: 4.93 SD: 0.95 NA: 0	Average: 5.03 SD: 1.07 NA: 0	*χ^2^* (1) = 0.46 *p* = 0.50
Prior adaptive Performance—Managing work stress	Average: 5.66 SD: 0.84 NA: 0	Average: 5.50 SD: 0.87 NA: 0	*χ^2^* (1) = 0.72 *p* = 0.40

Note. NA stands for Not Available and refers to missing values.

**Table 2 behavsci-13-00202-t002:** Sequences for each experimental condition and respective time.

Baseline	Soft Skills Metacognition Training
Measures pre-condition (self-efficacy, adaptive performance, metacognition of soft skills) 5 min	
Corporate meetings (usual professional activity) 1 h 30 to 2 h	Introduction 10 min
	Descriptive knowledge of soft skills 30 min
	Conditional and procedural knowledge of soft skills (Quiz) 15–25 min
	Deepening the 3 dimensions of metacognitive knowledge (autobiographical assessment and narrative about soft skills implementation) 35–50 min
Measures post-condition 5 min	

**Table 3 behavsci-13-00202-t003:** Descriptive statistics of the main variables.

Variables	n	M	SD	1	2	3	4	5	6	7	8	9	10	11	12	13	14
1. Soft skills Metacognition—Pre-condition	180	2.33	0.64	—													
2. Soft skills Metacognition—Post-condition	180	3.25	0.52	0.42	—												
3. Self-efficacy—Pre-condition	180	3.24	0.36	0.22	0.21	—											
4. Self-efficacy—Post-condition	180	3.35	0.35	0.19	0.37	0.70	—										
5. Adaptive performance—Dimension 1—Pre-condition	180	5.17	0.89	0.22	0.16	0.50	0.36	—									
6. Adaptive performance—Dimension 1—Post-condition	180	5.47	0.90	0.25	0.29	0.42	0.48	0.75	—								
7. Adaptive performance—Dimension 2—Pre-condition	180	5.23	0.80	0.20	0.18	0.45	0.45	0.45	0.35	—							
8. Adaptive performance—Dimension 2—Post-condition	180	5.62	0.75	0.05	0.26	0.42	0.58	0.28	0.42	0.70	—						
9. Adaptive performance—Dimension 3—Pre-condition	180	5.78	0.76	0.32	0.36	0.36	0.43	0.40	0.39	0.40	0.38	—					
10. Adaptive performance—Dimension 3—Post-condition	180	6.06	0.70	0.26	0.38	0.26	0.44	0.19	0.37	0.24	0.47	0.70	—				
11. Adaptive performance—Dimension 4—Pre-condition	180	5.01	1.05	0.37	0.30	0.25	0.25	0.37	0.41	0.37	0.32	0.51	0.39	—			
12. Adaptive performance—Dimension 4—Post-condition	180	5.34	0.97	0.18	0.29	0.15	0.27	0.32	0.44	0.31	0.41	0.35	0.43	0.77	—		
13. Adaptive performance—Dimension 5—Pre-condition	180	5.53	0.87	0.30	0.21	0.44	0.45	0.37	0.41	0.49	0.41	0.52	0.38	0.42	0.29	—	
14. Adaptive performance—Dimension 5—Post-condition	180	5.81	0.85	0.14	0.23	0.44	0.51	0.25	0.39	0.38	0.47	0.45	0.54	0.28	0.30	0.76	—

**Table 4 behavsci-13-00202-t004:** Outputs of models predicting post-condition adaptive performance sub-scores based on prior adaptive performance and experimental condition.

Adaptive Performance Dimension	Effect of Prior Adaptive Performance	Experimental Condition
Creative Problem Solving	***t*(173.48) = 15.98** ***b* = 0.77** ***p* < 0.001**	***t*(149.05)= 3.21** ***b* = 0.36** ***p* = 0.002**
Reactivity	***t*(176.96) = 13.85** ***b* = 0.68** ***p* < 0.001**	***t*(126.47)= 3.61** ***b* = 0.37** ***p* < 0.001**
Interpersonal Adaptability	***t*(175.98) = 13.57** ***b* = 0.66** ***p* < 0.001**	***t*(129.49)= 2.82** ***b* = 0.28** ***p* = 0.006**
Training and Learning	***t*(175.49) = 16.09** ***b* = 0.70** ***p* < 0.001**	***t*(116.79)= 2.66** ***b* = 0.32** ***p* = 0.009**
Working stress management	***t*(177.00) = 15.82** ***b* = 0.74** ***p* < 0.001**	*t*(146.20)= 1.84 *b* = 0.20 *p* = 0.07

Note. Experimental condition is dummy coded: 1 for soft skills metacognition training, 0 for Baseline. Tests in bold are significant.

**Table 5 behavsci-13-00202-t005:** Outputs of models predicting post-condition adaptive performance sub-scores based on pre-condition adaptive performance, pre- and post-condition soft skills metacognition, pre- and post-condition self-efficacy and experimental condition.

Adaptive Performance Dimension	Post-Condition Soft Skills Metacognition	Post-Condition Self-Efficacy	Experimental Condition
Creative Problem Solving	*t*(172.87) = 0.44 *b* = 0.05 *p* = 0.66	***t*(173.00) = 4.26** ***b* = 0.74** ***p* < 0.001**	*t*(155.30)= 1.32 *b* = 0.19 *p* = 0.19
Reactivity	*t*(172.71) = 0.63 *b* = 0.06 *p* = 0.53	***t*(173.00) = 4.99** ***b* = 0.77** ***p* < 0.001**	*t*(145.20)= 1.25 *b* = 0.16 *p* = 0.21
Interpersonal Adaptability	*t*(170.84) = 0.47 *b* = 0.05 *p* = 0.63	***t*(171.90) = 2.98** ***b* = 0.47** ***p* = 0.003**	*t*(156.93)= 1.07 *b* = 0.14 *p* = 0.29
Training and Learning	*t*(171.45) = 0.72 *b* = 0.09 *p* = 0.47	***t*(171.36) = 2.34** ***b* = 0.45** ***p* = 0.02**	*t*(146.57)= 0.69 *b* = 0.11 *p* = 0.49
Working stress management	*t*(170.94) = 0.82 *b* = 0.09 *p* = 0.41	***t*(171.23) = 2.75** ***b* = 0.46** ***p* = 0.007**	*t*(16.84)= 0.25 *b* = 0.03 *p* = 0.80

Note. Experimental condition is dummy coded: 1 for Soft skills metacognition training, 0 for Baseline. Tests in bold are significant.

## Data Availability

Data and analysis code are available at https://osf.io/3s4bm/files/ (accessed on 21 January 2023).

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
