# Peer review of "Effects of a New Soft Skills Metacognition Training Program on Self-Efficacy and Adaptive Performance"

_behavsci, 2023, doi:10.3390/bs13030202_

Round 1

Reviewer 1 Report

The research article submitted for review represents a clear scientific contribution. The review of the literature is adequate, collecting the position of different authors on the topic under study. The methodology and results are also consistent.

However, at a methodological level, it is recommended that the authors explain the limitations of using only subjects from the same organization. It would be advisable, for a greater representativeness of the results, to have collected information from organizations of different types (field of work, sector, company size, etc.).

The prospective of the study should be included in the conclusions.

Likewise, a consideration of the article “Transversal Competencies for Employability in University Graduates: A Systematic Review from the Employers’ Perspective” (https://doi.org/10.3390/educsci12030204) is recommended. This article follows a similar line.

Author Response

Response to Reviewer 1 Comments

Point 1: The research article submitted for review represents a clear scientific contribution. The review of the literature is adequate, collecting the position of different authors on the topic under study. The methodology and results are also consistent. However, at a methodological level, it is recommended that the authors explain the limitations of using only subjects from the same organization.

Response 1: Thank you very much for this relevant comment. We have chosen to focus our data collection on the population of a single company for practical reasons, which is that we had access to it. We are in the process of collecting data in other companies with the same experimental protocol, but we do not have yet enough data to include this new data in the analyses. You are right that having one company only constitutes a methodological limitation that we have not sufficiently highlighted in the first version of the manuscript. We have added in the discussion that further research is needed to check the generalizability of our findings, meaning that data have to be collected from different types of organizations.

Point 2: The prospective of the study should be included in the conclusions.

Response 2: Thank you for pointing this out. We have included a concluding paragraph in the article that outlines the perspectives of our research.

Point 3: Likewise, a consideration of the article “Transversal Competencies for Employability in University Graduates: A Systematic Review from the Employers’ Perspective” (https://doi.org/10.3390/educsci12030204) is recommended. This article follows a similar line.

Response 3: Thank you very much for this reference which relates our research object on several levels. We have added it to support the importance of soft skills as far as students’ employability is concerned.

Reviewer 2 Report

Introduction

It would be important to define what soft skills are when there are so many similar terms in the field, such as transferable skills, employability skills, 21st-century skills, and durable skills. In particular, since the manuscript refers to the "transferability" of soft skills, it would be important to include descriptions of how different "soft skills" are from "transferable skills". Such descriptions will help the authors justify the choice of the term “soft skills" especially when the term can give an impression that they are "weaker" than "hard skills," therefore, less important than "hard skills." 

Study Procedure

Although random effects were measured through the linear mixed model, it would be helpful to know more details about how the workshop facilitators were trained beforehand, what materials were used, how long the training lasted, etc.

Limitations

Although the authors recognize that soft skills are not like hard skills that can be acquired in a comparatively shorter time period, the experiment includes a single workshop that lasts about 2 hrs max which is a significant error in the design. There is a need to provide explanations for the choice of the time and frequency of the training. The authors mentioned a longitudinal study in Implications. The section should consider not only the lasting impact of the training but also a series of workshops in their future studies.   

The implications can be more practical and specific, such as how the present study contributes to the issue of reducing the gap between the skills employers require and the skills employees bring in?

Author Response

Response to Reviewer 2 Comments

Point 1: Introduction It would be important to define what soft skills are when there are so many similar terms in the field, such as transferable skills, employability skills, 21st-century skills, and durable skills. In particular, since the manuscript refers to the "transferability" of soft skills, it would be important to include descriptions of how different "soft skills" are from "transferable skills". Such descriptions will help the authors justify the choice of the term “soft skills" especially when the term can give an impression that they are "weaker" than "hard skills," therefore, less important than "hard skills." 

Response 1: Thank you for this comment which is in line with the work we carried out on a taxonomy of soft skills (Joie-La Marle et al., 2022). We have noted in this article that soft skills definition was quite blurry, all the more so because many terms are used as synonyms for soft skills, such as transferable skills (Goggin et al., 2019;  Suarta et al., 2017) although they are not equivalent. Transferable skills commonly refer to skills that are used in a large range of contexts, jobs and fields that encompass technical skills such as IT skills and numeracy (Roskosa & Stukalina, 2017, Hyland, 2006).  Nevertheless, soft skills are defined as non-technical since the emergence of this term (Whitmore, 1972) and this characteristic is remarkably consensual (Hurrell, 2016; Matteson et al., 2016; Nasir et al., 2011). This non-technical characteristic, as well as the fact that soft skills are acquired through experience (Chell & Athayde, 2011; Nitonde, 2014), can be executed without awareness (Mauleon et al, 2014) and are not formally taught  (Haseeb et al., 2021)  is particularly interesting as far as potential metacognition impact is concerned. That’s why we chose to focus on the term “soft skills” in this study.  

As far as the definition of soft skills is concerned, we have limited ourselves to the most consensual characteristics in the literature, which are the fact that they are non-technical and transversal

We included in the article details on the distinction between transferable skills and soft skills and insisted on the non-technical dimension in their definition.

Reference non included in the article:

Haseeb, M., Azfar, M. W., Ahmed, M., Tariq, A., Nawaz, M. S., & Sadiq, A. (2021). Development and validation of scale for self evaluation of soft skills in postgraduate dental students. JPMA. The Journal of the Pakistan Medical Association71(1 (Suppl 1)), S9.

Point 2: Study Procedure Although random effects were measured through the linear mixed model, it would be helpful to know more details about how the workshop facilitators were trained beforehand, what materials were used, how long the training lasted, etc.

Response 2: Thank you for your request for clarification. We have added information in the manuscript about the training of facilitators. Indeed, we have standardized the delivery of our 2-hour workshops by implementing a training process for our facilitators.

Specifically, the training of facilitators is governed by a 4-step process:

  1. first the future facilitator takes part in the workshop as a participant
  2. then he/she attends the workshop as an observer of a senior facilitator
  3. then he/she participates in the animation of the workshop paired with a senior facilitator
  4. finally, he/she conducts the workshop under the supervision of a senior facilitator

We also added in the article precisions on the material used during the workshop.

This material includes of a booklet defining the soft skills discussed, and cards showing these definitions and displayed on a tray to facilitate pedagogical visualization by all participants at any time during the workshop. These soft skills cards are also used to support the participants' vote at the end of the workshop for the soft skills identified in their colleagues' pitch: tokens are provided to the players so that they can carry out this vote by placing them on the different cards. The workshop material also includes quiz cards that are distributed in the first phase of soft skills recognition.

Point 3: Limitations Although the authors recognize that soft skills are not like hard skills that can be acquired in a comparatively shorter time period, the experiment includes a single workshop that lasts about 2 hrs max which is a significant error in the design. There is a need to provide explanations for the choice of the time and frequency of the training. The authors mentioned a longitudinal study in Implications. The section should consider not only the lasting impact of the training but also a series of workshops in their future studies.   

Response 3: Thank you very much for your relevant comment. We explained in the article the reasons that led us to choose a protocol based on a single 2-hour workshop. As this protocol was not optimal to obtain an optimal impact of soft skills metacognition, we explained this limitation and proposed to complement this work with research including a methodology based on a series of workshops and not a single workshop.

Point 4: The implications can be more practical and specific, such as how the present study contributes to the issue of reducing the gap between the skills employers require and the skills employees bring in?

Response 4: Thank you for pointing this out. We have completed the paragraph on the implications of our research with more practical considerations, particularly concerning the reduction of the soft skills gap.